# PHRASETRANSFORMER: SELF-ATTENTION USING LOCAL CONTEXT FOR SEMANTIC PARSING

## ABSTRACT

Semantic parsing is a challenging task whose purpose is to convert a natural language utterance to machine-understandable information representation. Recently, solutions using Neural Machine Translation have achieved many promising results, especially Transformer because of the ability to learn long-range word dependencies. However, the one drawback of adapting the original Transformer to the semantic parsing is the lack of detail in expressing the information of sentences. Therefore, this work proposes a PhraseTransformer architecture that is capable of a more detailed meaning representation by learning the phrase dependencies in the sentence. The main idea is to incorporate Long Short-Term Memory (LSTM) into the Self-Attention mechanism of the original Transformer to capture more local context of phrases. Experimental results show that the proposed model captures the detailed meaning better than Transformer, raises local context awareness and achieves strong competitive performance on Geo, MSParS datasets, and leads to SOTA performance on Atis dataset in methods using Neural Network.

## 1 INTRODUCTION

Semantic parsing is an important task which can be applied for many applications such as Question and Answering systems or searching systems using natural language (Woods, 1973; Waltz & Goodman, 1977). For example, the sentence *"which state borders hawaii"* can be represented as logical form (LF) using $\lambda$-calculus syntax *"(lambda $0 e (and (state:t $0) (next_to:t $0 hawaii)))"*. There are various strategies to address the semantic parsing task such as constructing handcraft-rules (Woods, 1973; Waltz & Goodman, 1977; Hendrix et al., 1978), using Combinatory Categorial Grammar (CCG) (Zettlemoyer & Collins, 2005; 2007; Kwiatkowski et al., 2011), adapting statistical machine translation method (Wong & Mooney, 2006; 2007) or Neural Machine Translation (Dong & Lapata, 2016; Jia & Liang, 2016; Dong & Lapata, 2018; Cao et al., 2019). The major factor of the CCG method is based on the alignments of sub-parts (lexicons or phrases) between a natural sentence and corresponding logical form and to learn how best to combine these subparts. In more detail, the phrase *"borders hawaii"* is aligned to *"(next_to:t $0 hawaiiz)"* in LF. Conversely, the methods using Neural Machine Translation learn the encoder representing a sentence into a vector and decode that vector into LF. The current SOTA models are Sequence-to-Sequence using LSTM (Seq2seq) (Dong & Lapata, 2018; Cao et al., 2019) on Geo, Atis and Transformer (Ge et al., 2019) on MSParS. The methods using Neural Network almost work effectively without any handcrafted features. However, there is still room to improve the performance based on the meaning of local context in phrases.

According to CCG methods, the semantic representation of a sentence is the combination of sub-meaning representation generated by phrases in a sentence. However, Transformer architecture

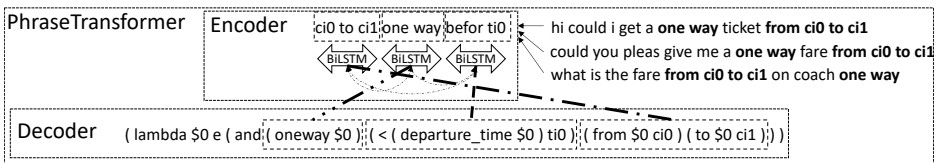

Figure 1: Pharse alignments in PhraseTransformer.

only learns the dependencies between single words without considering the local context by the phrase. Therefore, we propose a new architecture named PhraseTransformer that focuses on learning the relations of phrases in a sentence (Figure 1). To do this, we modify the Multi-head Attention (Vaswani et al., 2017) by applying the self-attention mechanism into phrases instead of single words. Firstly, we use $n$-gram to split a sentence into phrases. Then, we use the final hidden state of LSTM architecture to represent the local context meaning of those phrases.

Our contributions are: (1) proposing a novel model based on Transformer that works effectively for semantic parsing tasks, (2) conducting experiments to confirm the awareness capacity of the model, (3) achieving competitive performance on Geo, MSParS datasets and new SOTA performance on Atis dataset in the methods using Neural Network.

## 2 RELATED WORK

In **Semantic Parsing** task, recent works have shown that using the deep learning approach achieved potential results. results. These methods are divided into three groups:

*Decoder Customization.* Dong & Lapata apply the Seq2seq model to semantic parsing task and introduce Sequence-to-tree (Seq2tree) (Dong & Lapata, 2016) model constructing the tree structure of the LF. This model focuses on modifying the decoding method based on bracket pairs to start a new decoding level. On an other aspect, Dong & Lapata (2018) continue to introduce a new architecture Coarse-to-Fine (Coarse2Fine) based on a rough sketch of meaning to improve the structure-awareness of Seq2seq model. Similarly, Li et al. (2019) also use the sketch meaning mechanism on BERT model (Devlin et al., 2019) by two steps: classify the template of LF and fill the low-level information to that template. In our opinion, the main problem is to improve the understanding capacity of the model because semantic parsers need to capture the complicated in the natural sentences before decoding. Therefore, our work focuses on designing the Encoder architecture to improve the understanding capacity of the model.

*Data Augmentation.* There are numerous works that focus on data augmentation to improve the performance of the semantic parsing model (Jia & Liang, 2016; Ziai, 2019; Herzig & Berant, 2019). Jia & Liang propose three rules based on Synchronous Context-Free Grammar to recombine data. This step increases the size of the training data and grows the performance of the model (Jia & Liang, 2016). Similarly, Ziai proposes a method that automatically augments data based on the co-occurrence of words in the sentence. The author separates the training process into two phases: (1) use augmented data to train for BERT (Devlin et al., 2019) and (2) fine-tuning on original data.

*Weak Supervision.* Some methods use semi-supervised learning for semantic parsing task such as (Kočiský et al., 2016; Yin et al., 2018; Goldman et al., 2018; Cao et al., 2019; 2020). These works are promising approaches for the data-hungry problem because of the ability to extract latent information such as unpaired logical forms. In our proposed model, we aim to construct the latent representation for phrases and learn these representations via the self-attention mechanism of the Transformer. We hypothesize that complicated sentences are constructed from various phrases, so learning to represent these phrases makes the model more generalizable.

In **Neural Machine Translation** task, the approach using phrase information or constituent tree is proved that effective and attracts many works (Wang et al., 2017; Wu et al., 2018; Wang et al., 2019; Hao et al., 2019; Nguyen et al., 2020). The points that make the difference in our work are: (1) our model is capable of learning without any additional information (e.g. constituent tree), (2) in the training process, although we do not force the attention or limit the scope of the dependencies, our model is able to pay high attention to the important phrase automatically. Compare with Yang et al. (2018), the purpose of using local context information is similar but different in *localness modeling*: based on the distance, Yang et al. (2018) cast a Gaussian bias to change attention score while our method is simpler by incorporating multi different n-gram views as the various local contexts.

## 3 MODEL ARCHITECTURE

Our novel architecture (Figure 2) is based on the Encoder-Decoder of Transformer (Vaswani et al., 2017). We define a new model named *PhraseTransformer* to improve the encoding quality of Transformer by enhancing the Encoder architecture while keeping the original Decoder.

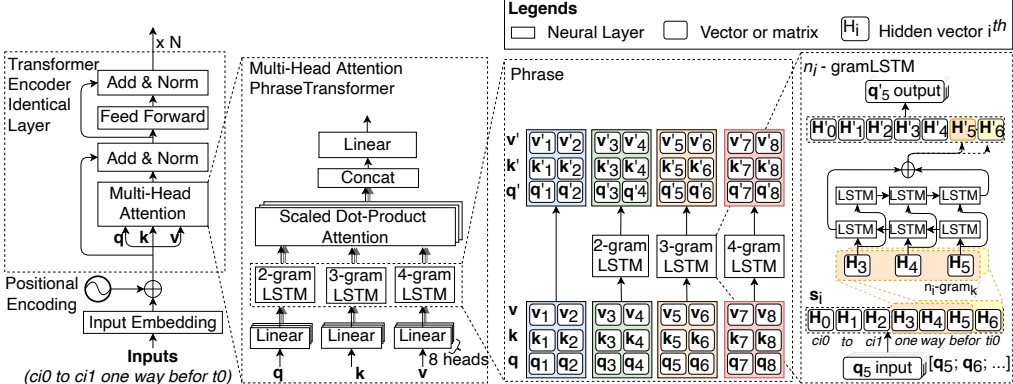

Figure 2: PhraseTransformer Encoder architecture using $n$-gram LSTM in MultiHead Layer. In this case, $n$-gramLSTM layer is built with $\boldsymbol{n} = [0, 0, 2, 2, 3, 3, 4, 4]$, 2-gram, 3-gram, 4-gram models apply to every two heads from head 3 to head 8.

**Transformer Encoder (original).** In Transformer Encoder architecture, Vaswani et al. (2017) proposed a stack of $N$ Identical Layers; each layer consists of two sub-layers: Multi-Head Attention layer and Position-wise Feed-Forward layer. Let $\boldsymbol{x}$ be an input vector synthesized from the vector word embedding and positional encoding $\boldsymbol{x} = [\boldsymbol{x}_1, ..., \boldsymbol{x}_{|S|}]$ where $|S|$ is sentence length.

In the Multi-Head Attention layer Vaswani et al. use the Linear layer to get multi-views for the inputs. This layer processes the input vector ($\boldsymbol{x}$) and generates $H$ distinct featured vectors ($H$ is the number of heads) and forwards to Self-Attention layer using Scaled-Dot Product. After that, all heads are processed by Concat and Linear layers to compute the output of the Multi-Head layer.

$$\boldsymbol{q}_i, \boldsymbol{k}_i, \boldsymbol{v}_i = \boldsymbol{x}\boldsymbol{W}_i^q, \boldsymbol{x}\boldsymbol{W}_i^k, \boldsymbol{x}\boldsymbol{W}_i^v \tag{1}$$

$$\boldsymbol{head}_i = \text{Attention}(\boldsymbol{q}_i, \boldsymbol{k}_i, \boldsymbol{v}_i) \tag{2}$$

$$\boldsymbol{h}_{MulH} = [\boldsymbol{head}_1; ...; \boldsymbol{head}_H]\boldsymbol{W}^o \tag{3}$$

$$\boldsymbol{h}_{Norm} = \text{LayerNorm}(\boldsymbol{h}_{MulH} + \boldsymbol{x}) \tag{4}$$

$$\boldsymbol{h}_{out} = \text{LayerNorm}(\text{FeedForward}(\boldsymbol{h}_{Norm}) + \boldsymbol{h}_{Norm}) \tag{5}$$

where Attention is Scaled Dot-Product Attention:

$$\text{Attention}(\boldsymbol{q}_i, \boldsymbol{k}_i, \boldsymbol{v}_i) = \text{softmax}(\frac{\boldsymbol{q}_i \boldsymbol{k}_i^{\mathsf{T}}}{\sqrt{d_h}})\boldsymbol{v}_i \tag{6}$$

where $d_h$ is dimensions per head, $i$ is the identical index of head ($0 < i \leq H$), $\boldsymbol{W}$ is parameters, LayerNorm, FeedForward are the functions that are used similar to Vaswani et al. (2017).

**PhraseTransformer Encoder.** The Encoder is enhanced from the original model in Multi-Head Attention layer because this layer is the major factor to extract the features of inputs sequence. More detail, after $H$ heads are generated by Linear layer, we use $n$-gram model to split the sentence into grams and use Bidirectional LSTM (Hochreiter & Schmidhuber, 1997) to extract the local context information of these grams (Figure 2). Besides, we assume that the meaning phrases are usually composed by difference length, therefore we use various $n$-gram models. To do this, the Phrase function is in Equation 7:

$$\text{Phrase}(\boldsymbol{s}_i) = \begin{cases} n_i\text{-gramLSTM}(\boldsymbol{s}_i) & \text{if} \quad n_i \neq 0 \\ \boldsymbol{s}_i & \text{otherwise} \end{cases} \tag{7}$$

where $\boldsymbol{s}_i$ is a sequential hidden state of a sentence of head $i$ ($0 < i \leq H$) in Multi-Head layer; $\boldsymbol{n} \in \mathbb{N}^H$ is gram size vector for $H$ heads; $n_i$ is the gram size corresponding to head $i$; $n_i$-gramLSTM is a procedure that splits the sequential input into grams by $n_i$-gram model, and applies Bidirectional LSTM for each gram $k$ of $\boldsymbol{s}_i$:

$$n_i\text{-gramLSTM}(\boldsymbol{s}_i) = [n_i\text{-gramLSTM}_k(\boldsymbol{s}_i)] \tag{8}$$

where $n_i\text{-gramLSTM}_k$ is the Bidirectional LSTM computed by sum of forward and backward final hidden state:

$$n_i\text{-gramLSTM}_k(\boldsymbol{s}_i) = LSTM_i^f(n_i\text{-gram}_k(\boldsymbol{s}_i)) + LSTM_i^b(n_i\text{-gram}_k(\boldsymbol{s}_i)) \tag{9}$$

$$n_i\text{-gram}_k(\boldsymbol{s}_i) = [\boldsymbol{H}_{k-n_i+1}; \boldsymbol{H}_{k-n_i+2}; ...; \boldsymbol{H}_k] \tag{10}$$

where $\boldsymbol{H}_k$ is the hidden state corresponding to word index $k$ in a sentence, $n_i\text{-gram}_k$ is the gram index $k$ that is a list of $n_i$ continuous hidden states (paddings zero for first words), $n_i\text{-gramLSTM}_k(\boldsymbol{s}_i)$ is the vector to capture local context of the gram index $k$. After that, the query ($\boldsymbol{q}_i$), key ($\boldsymbol{k}_i$), value ($\boldsymbol{v}_i$) matrixes (Equation 2) are replaced by *Phrase* function:

$$\boldsymbol{q}_i', \boldsymbol{k}_i', \boldsymbol{v}_i' = \text{Phrase}(\boldsymbol{q}_i), \text{Phrase}(\boldsymbol{k}_i), \text{Phrase}(\boldsymbol{v}_i) \tag{11}$$

$$\boldsymbol{head}_i = \text{Attention}(\boldsymbol{q}_i', \boldsymbol{k}_i', \boldsymbol{v}_i') \tag{12}$$

*Residual Connection.* Similar to the original Transformer architecture, we also employ a residual connection as an extension aim to effectively integrate local context features and current word information. We used $\text{sigmoid}(\sigma)$ function to adjust the rate of context ($n_i\text{-gramLSTM}_k(\boldsymbol{s}_i)$) and current word ($\boldsymbol{s}_{i,k}$) information. The hidden state $n_i\text{-gramLSTM}_k$ in Equation 8 is replaced as following:

$$n_i\text{-gramLSTM}_k'(\boldsymbol{s}_i) = \sigma(\boldsymbol{s}_{i,k}) \cdot n_i\text{-gramLSTM}_k(\boldsymbol{s}_i) + (1 - \sigma(\boldsymbol{s}_{i,k})) \cdot \boldsymbol{s}_{i,k} \tag{13}$$

Finally, $\boldsymbol{h}_{MulH}, \boldsymbol{h}_{Norm}, \boldsymbol{h}_{out}$ are computed similarly to Transformer architecture.

**Model variantion.** We replace the method representing local context (Bidirectional LSTM) by an other simple method that is *Sum* of all hidden state of words in the phrase. More detail, we customize the *Phrase* function by replacing $n\text{-gramLSTM}$ (Equation 7) with $n_i\text{-gramSum}$:

$$n_i\text{-gramSum}(\boldsymbol{s}_i) = [n_i\text{-gramSum}_k(\boldsymbol{s}_i)] \tag{14}$$

$$n_i\text{-gramSum}_k(\boldsymbol{s}_i) = \sum(n_i\text{-gram}_k(\boldsymbol{s}_i)) \tag{15}$$

where $n_i\text{-gram}_k(\boldsymbol{s}_i)$ function is computed similar to the Equation 10.

**Training method** is to maximize the Log-Likelihood function of the probabilities to generate the LF ($y$) given a sentence ($x$) from annotated dataset ($\mathcal{D}$):

$$\text{maximize}: \sum_{<x,y>\in\mathcal{D}} \log p_\theta(y|x) \tag{16}$$

**Metric measurement.** On all datasets, we compute sentence-level accuracy by using exact matching (EM) and logic matching (LM) that developed by Dong & Lapata (2018). LM metric measures the performance better than the EM method because it is probable for comparing the variant of expression. For example, the predicted LFs in different order of *and* logic: *and ( oneway $0 ) ( <( departure_time $0 ) ti0 )* is equal to *and ( <( departure_time $0 ) ti0 ) ( oneway $0 )* .

## 4 EXPERIMENTS

The purpose of experiments is to compare the performance of PhraseTransformer and extension models with the original Transformer. Besides, we explore the awareness about the phrase alignment between a sentence and the generated LF by PhraseTransformer.

### 4.1 DATATSETS

We conduct the experiments on three datasets Geo (Zelle & Mooney, 1996), Atis (Dahl et al., 1994) and MSParS (Duan, 2019). Table 1 shows the observation of these datasets. Geo and Atis datasets are small size but more complicated in information relations than the MSParS dataset. The average length of LFs on Atis dataset (28.4) is about twice longer than that on MSParS dataset (14.7). The original MSParS dataset have large vocabulary (around 40k) because it consists of various entities name in the open domain. Therefore, we preprocess this dataset similarly to Ge et al. (2019) by replacing character "_" by " _ " and using byte-pairs-encoding (BPE) (Sennrich et al., 2016) to deal with rare-word problem.

- **Geo** consists of queries about geography information of the U.S. and LFs in lambda-calculus syntax. We use the version preprocessed by Dong & Lapata (2016) by replacing all entities by numbered markers (e.g. *"new york" → "s0"*).

- **Atis** consists of queries about flight information and LFs in lambda-calculus syntax. We also use the version preprocessed by Dong & Lapata (2016) similar to Geo dataset.

- **MSParS** is a large-scale open domain dataset with LFs in lambda-DCS syntax (Liang et al., 2011). This dataset contains 12 question types (Duan, 2019) such as single-relation, multi-turn-entity, etc. for Knowledge-based Question Answering system.

Table 1: Statistics information of three datasets. The MSParS dataset (BPE6k) is preprocessed by BPE 6000 operations. Vocabulary size and average length of source (Src) and target (Tgt) side are computed on train set.

| Datatset | Total examples Train - Dev - Test | Vocab size Src | Tgt | Avg. length Src | Tgt |
|---|---|---|---|---|---|
| Geo | 600 - 0 - 280 | 433 | 51 | 10.6 | 18.7 |
| Atis | 3434 - 491 - 448 | 120 | 166 | 7.3 | 28.4 |
| MSParS (BPE6k) | 63826 - 9000 - 9000 | 4965 | 5854 | 12.8 | 23.9 |

## 4.2 SETTINGS

In training processes of the MSParS and Atis datasets, to prevent overfitting, we use the *early stopping* conditioned on metrics word-level or sentence-level accuracy dev set.

**Hyper-parameters.** Because Transformer is quite sensitive in hyper-parameters, we keep most hyper-parameters the same as Transformer-base model (Vaswani et al., 2017) such as the number of layers $N = 6$ and number of heads in Multi-Head layer is $H = 8$; hidden size $d_{model} = 512$; dropout is selected in $\{0.1, 0, 3\}$; Adam optimizer with $\beta_1 = 0.9, \beta_2 = 0.998, \epsilon = 10^{-9}$. The weights of models are initialized with Xavier initialization (Glorot & Bengio, 2010). The embedding vectors are shared among the source-side and target-side, between the input-to-embedding layer and output-to-softmax layer in Decoder. We also retain the learning rate decay method: $lr(step) = d_{model}^{-0.5} \cdot \min(step^{-0.5}, step \cdot warmup\_steps^{-1.5})$ where $step$ is the current step number. The $n$-gram size for each head is selected in $\{0, 2, 3, 4\}$. The weights of Bidirectional LSTM layers in the heads using the same $n$-gram model (e.g. heads 3, 4) are shared. Besides, the experimental dataset sizes are quite different, therefore we use three hyper-parameter sets[1]: **Geo**: $warmup\_step = 100$ learning rate init selected from $\{\mathbf{0.05}, 0.1\}$, $batch\_size = 128$ (the batch size using number of tokens), the maximum traning steps $max\_steps = 15000$; **Atis**: $warmup\_step = 100$ learning rate init selected from $\{\mathbf{0.1}, 0.2\}$, $batch\_size = 4096$. the maximum traning steps $max\_steps = 250000$; **MSParS**: $warmup\_step = 8000$, learning rate init selected from $\{0.5, 1.0, \mathbf{2.0}\}$, $batch\_size = 8192$, the maximum traning steps $max\_steps = 250000$. On this dataset, we conducted experiments to check the number of BPE operations impacting to performance (Figure 3). Based on those results, we use the MSParS dataset preprocessed by BPE 6000 operations for all other experiments.

## 4.3 RESULTS AND ANALYSIS

### 4.3.1 PERFORMANCE

**Model setting** We conducted experiments to find the best gram sizes for PhraseTransformer on Atis and MSParS (Table 2) because the size of those datasets are larger than Geo that make the results are more stable. We hypothesize that performance increases when applying various gram sizes to the Atis dataset. By using various gram sizes, PhraseTransformer can see different linguistic features in various local context sizes in Multi-head layers. For domain adaptation, the gram sizes can be chosen depending on observing the number of words in meaningful phrases. Using various gram sizes makes PhraseTransformer more generalized. Besides, using LSTM to represent spans on all layers helps PhraseTransformer capture more sequential information than Transformer.

---

[1]The model using bold value is achieved a better performance than other values in our experiments.

*Residual Connection.* On MSParS dataset, the performance is not so different when changing gram sizes (models 1 - 4 Table 2). We observe that because this dataset has diversity in object name with more than 75% words in vocabulary appearing less than 4 times in train set. One of the challenge of this dataset is to recognize the object name and type, so capturing original word features is important. These words are usually splited into many word pieces by BPE, so the $n_i$_gramLSTM component lose original word information when intergrating parts of previous word. For example, the sentence *"boonie bears last movie was"* is preprocessed by BPE: *"bo@@ on@@ ie be@@ ars last movie was"* and the $n_i$_gramLSTM component considers similar grams *[bo@@ on@@ ie be@@], [on@@ ie be@@], [ie be@@]* when representing *be@@* word vector. PhraseTransformer equipped with the residual connection (model 5) is able to avoid losing original word pieces features, thus shows better performance on MSParS.

*Model variantion.* We conducted the experiments to check the impact of localness modeling between BiLSTM (model 5) and the Sum function (model 6). On both datasets, the model using BiLSTM achieved better performance because the LSTM model is better than the Sum function in meaning representation. However, on the Atis dataset, PhraseTransformer using Sum improved slightly (about 0.3 %) with the original Transformer. This result shows that local context is one of the important features for this dataset.

Table 2: Sentence-level accuracy using exact matching (EM) and logic matching (LM) on two datasets Atis and MSParS using BPE 6000 operations. The abbreviation Res. implies that we used residual connection in Equation 13.

| Id. Model | gram sizes ($n$) | Atis (EM/LM) | | MSParS (EM/LM) | |
|---|---|---|---|---|---|
| | | Dev | Test | Dev | Test |
| 1. PhraseTrans. | $[0;0;0;0;2;2;2;2]$ | 86.76 / 88.80 | 87.95 / 88.84 | 85.62 / 85.99 | 84.68 / 85.18 |
| 2. PhraseTrans. | $[0;0;0;0;3;3;3;3]$ | 86.76 / 88.80 | 88.17 / 89.51 | 86.07 / 86.52 | 85.13 / 85.72 |
| 3. PhraseTrans. | $[0;0;0;0;2;2;3;3]$ | 86.76 / 88.19 | 89.06 / 89.96 | 85.53 / 85.99 | 85.04 / 85.39 |
| 4. PhraseTrans. | $[0;0;2;2;3;3;4;4]$ | 87.17 / 89.21 | **89.51 / 90.40** | 85.88 / 86.24 | 85.08 / 85.47 |
| 5. PhraseTrans.Res. | $[0;0;2;2;3;3;4;4]$ | **87.58 / 89.61** | 88.62 / 89.51 | **86.23 / 86.73** | **85.72 / 86.21** |
| 6. PhraseTrans.Sum | $[0;0;2;2;3;3;4;4]$ | 86.96 / 88.59 | 87.05 / 87.95 | 85.68 / 86.18 | 85.21 / 85.82 |

Table 3: Evaluation results using Logic Matching on all datasets. The reported results on Geo are mean and standard deviation values. The values marked (*) mean that the evaluation metric is denotation match that different from others using sentence-level accuracy. This table contains two parts, the upper part shows the results of previous works and bellow part present our results. Models 4, 5 refer the Id of model in Table 2.

| | Geo | Atis | MSParS |
|---|---|---|---|
| Z&C (Zettlemoyer & Collins, 2007) | 86.1 | 84.6 | |
| λ-WASP (Wong & Mooney, 2007) | 86.6 | | |
| FUBL (Kwiatkowski et al., 2011) | 88.6 | 82.8 | |
| TISP (Zhao & Huang, 2015) | **88.9** | 84.2 | |
| Seq2tree (Dong & Lapata, 2016) | 87.1 | 84.6 | |
| Seq2seq+Copy (Jia & Liang, 2016) | 89.3* | 83.3 | |
| Coarse2Fine (Dong & Lapata, 2018) | 88.2 | 87.7 | |
| DualLearning (Cao et al., 2019) | | 89.1 | |
| Bert-Sketch (Li et al., 2019) | | | 84.47 |
| Transformer (Ge et al., 2019) | | | 85.68 |
| Transformer (ours) | 86.8±0.76 | 87.7 | 86.19 |
| PhraseTrans. (Model 4) | 87.9±0.36 | **90.4** | 85.47 |
| PhraseTrans.Res. (Model 5) | | 89.5 | **86.21** |

**Other methods comparison** We compare the performance of PhraseTransformer with the original Transformer and other methods in previous works in Table 3. The learning curve in Figure 4 also shows that PhraseTransformer beat clearly Transformer on the Geo dataset on all checkpoints. On Atis dataset, PhraseTransformer is better than Transformer on all settings of gram sizes (Table 2).

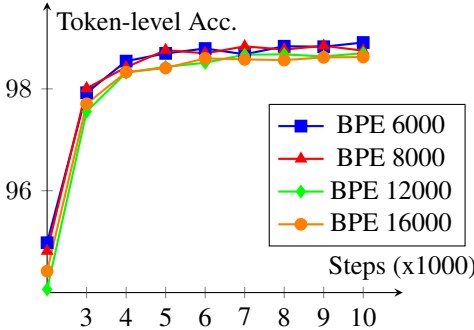
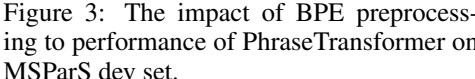
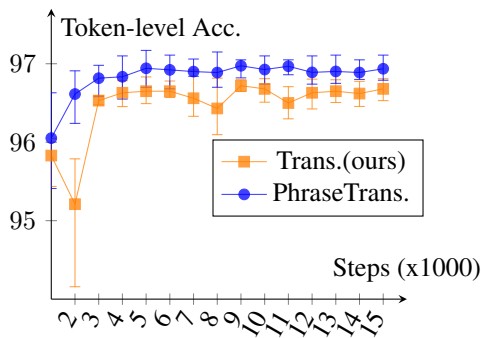

Figure 3: The impact of BPE preprocessing to performance of PhraseTransformer on MSParS dev set.

Figure 4: Token-level accuracy (min, max and average) of PhraseTransformer and the original Transformer on Geo test set.

Our model achieves better results on Atis, MSParS and so competitive with previous results on the Geo dataset. While our method does not use augmented datasets similarly to Jia & Liang (2016); Ge et al. (2019) or the sketch information (Dong & Lapata, 2018), these results show that our model learns more effectively than the others.

**PhraseTransformer Encoder layers**  Some recent works (Hao et al., 2019; Yang et al., 2018) show that the different combinations of the layers caturing local context can impact the performance of the model. Therefore, we conducted the experiments that drop the phrase mechanism on some top layers to explore this impact (Table 4). Comparing with the original Transformer, the PhraseTransformer improved performance even if only applying phrase mechanism on the first layer. Besides, PhraseTransformer is more general when applying phrase mechanism on all layers.

Table 4: Evaluation using Logic Matching for PhraseTransformer in difference layers on Atis dataset. Column *Layers* indicates the layers applying $n\_gramLSTM$.

| Layers | #Param. | Dev | Test |
|--------|---------|-------|-------|
| $[1]$ | 44.5 M | 89.21 | 88.17 |
| $[1-2]$ | 44.7 M | 88.80 | 89.06 |
| $[1-3]$ | 44.9 M | **89.41** | 89.29 |
| $[1-6]$ | 45.5 M | 89.21 | **90.40** |
| $[3-6]$ | 44.9 M | 89.00 | 89.29 |

Table 5: Comparison of number parameters (M=million) and training speed (tokens per second, K=thousand) on MSParS dataset.

| Model | #Param. | Speed |
|-------|---------|-------|
| Transformer (ours) | 47.1 M | 9.0 K |
| PhraseTrans. | 48.3 M | 7.1 K |
| PhraseTrans.Res. | 48.3 M | 6.9 K |

**Computation time**  We compare the number of parameters and training speed between the Transformer and PhraseTransformer on the largest dataset - MSParS (Table 5). This experiment is conducted on 1 GPU P100, 16Gb ram with batch size is 8192 tokens. The training speed of PhraseTransformer model is about 76-79% of the original Transformer. In fact, although we used LSTM on Heads, the computation time is not dependent on the length of sentence because we can forward and backward all $n$-grams of all sentences in a minibatch at the same time. Therefore, the computation time is more dependent on the gram size (in this case, the maximum gram size is 4). Besides, the number of parameters of the PhraseTransformer is slightly increased (about 2.5%) when compare with the original Transformer.

### 4.3.2 SELF-AWARENESS

**Alignment**  We inspect the information learned in PhraseTransformer in Attention layers (Figure 5a, more in Appendix A.2). We observe that PhraseTransformer could represent attention information more clearly than Transformer. In both two models, the token *ground_transport* in LF is aligned correctly to phrase *"ground transport"* in the sentence (red alignments). In PhraseTransformer, tokens *to_city*, *from_airport* are also correctly aligned to the corresponding words *"ap0"*,

*"ci0"* in the sentence (green and yellow alignments) because these word vectors probable to capture local context better than Transformer. Besides, all tokens decoded by PhraseTransformer paid the same attention to other words that is not key information, such as *"is there"*, *"into"*, *"citi"*. These evidences is positive signals show that the self-awareness of PhraseTrans better than Transformer.

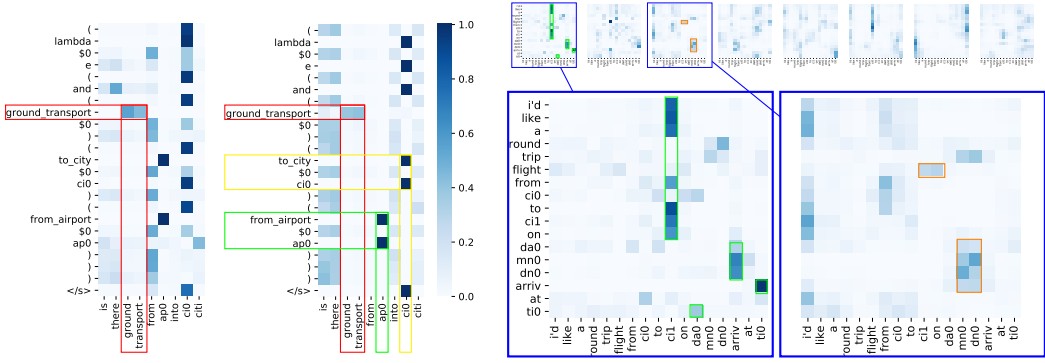

(a) Encoder-Decoder Attention.      (b) Self-Attention of PhraseTransformer Encoder.

Figure 5: Heatmap visualization of Attention. Figure a shows the difference of alignment Encoder-Decoder attention between the original Transformer (left) and PhraseTransformer (right). Considering one row, the value in each column is corresponding to the rate of the attention of token in LF to the word in the sentence. Figure b shows Self-Attention in 8 heads of the last PhraseTransformer Encoder layer. Two blue rectangles are zoomed-in separately of head 1 (not use $n$_gramLSTM), head 3 (use 2_gramLSTM).

Figure 5b (bigger version in Appendix A.1) shows the difference between heads in Self-Attention Encoder of PhraseTransformer. The self-attention in heads that do not use $n$_gramLSTM is more incoherent than other heads. For example, in head 1, almost words in query focus on *"ci1"* and the other words are paid attention is key information words such as *"da0", "arriv", "ti0"* (the green rectangles). From head 3 to 8, the attention focuses on the separated clusters, which shows that model learned the dependencies of the phrases instead of the single words. On these heads, the attentions are usually between groups important words such as *"flight"* with *"ci1 on"*, *"da0 nm0 dn0 arriv"* with *"nm0 dn0"* (the orange rectangles).

**Meaning phrase**    In this experiment, we explore the natural language understanding capacity of our PhraseTransformer. We use Principal Component Analysis (PCA) method to visualize the similarity of phrases in PhraseTransformer in Figure 9 by using hidden state of heads 7, 8 (the vector $[q'_7; q'_8]$ where $q'_i$ from Equation 11). We also highlight 30 closest points (the distance using Cosine distance) to the particular phrase carrying key information such as *"round trip"*, *"from ci1 to ci0"*. Besides, we also visualize the vector of words ($[q_7; q_8]$ where $q_i$ from Equation 1) to show the lacked local context information of word vectors in the original Transformer in Appendix A.3.

Considering two phrases *"from ci1 to ci0"* and *"from ci0 to ci1"* in Figure 6a, the phrases closest to two phrases concentrate on blue and cyan clusters. These two clusters are closest to each other but separate without overlapping. This feature helps the decoder decode different semantic components such as *(from $0 ci0) (to $ 0 ci1)* and *(from $0 ci1) (to $0 ci0)*. Figure 6b shows that the phrase *"from ci1 to ci0"* is represented by the similar vectors in various contexts. For example, this phrase in Atis data sentence 175 *"show me nonstop flight from ci1 to ci0"* has the same meaning in sentence 339 *"a flight from ci1 to ci0 arriv between ti0 and ti1"*. In Figure 6c, there are many different phrases have the same meaning that the model finds out, such as *"could i have"*, *"tell me again"*, *"find me all"* or the phrases closest to *"list the"* and *"show me all"* in Figure 6a. These phrases do not consist of query information, which is the robustness feature of human natural language, this is evidence that the model is capable of learning complicated characteristics of natural language.

**Examples of improvement**    We analyze examples that our PhraseTransformer improved over the original Transformer (Table 6). The improvement can be grouped into three types of errors: (1) 46.2% the errors are caused by Transformer confusing the role of entities name such as *"ci2"* and

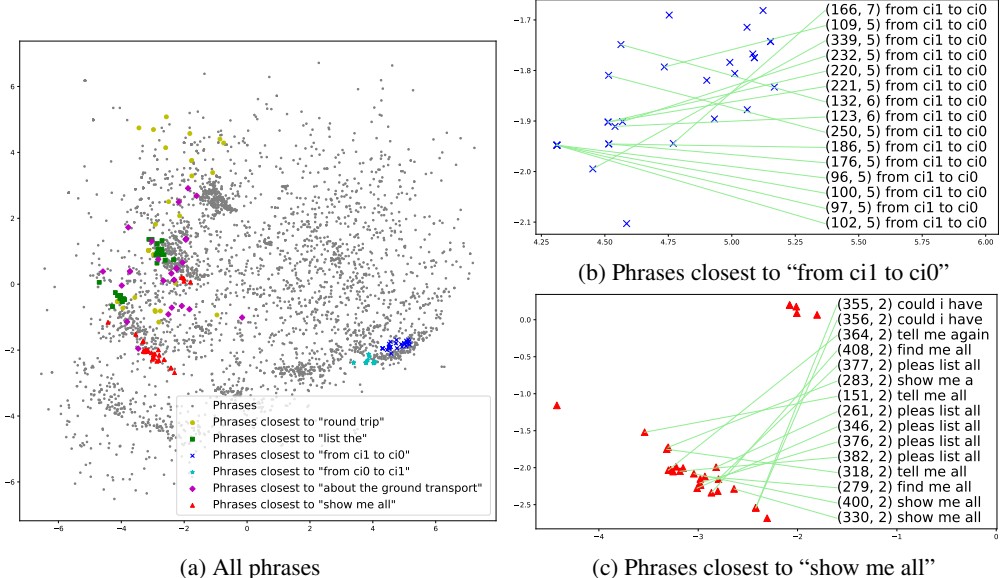

(a) All phrases     (b) Phrases closest to "from ci1 to ci0"

(c) Phrases closest to "show me all"

Figure 6: Figure a draws the representing vector of phrases in Selft-Attention Layer using PCA on Atis test set. Figures b, c are zoomed-in view of the blue and red clusters. The labels are annotated for each point in two figures show the information of the phrase corresponding to point following the template *(sentence_id, pharse_position) phrase_content*.

*"ci0"* (Row 1 on Table 6); (2) 27.3% missing semantic components such as *"(round_trip $0)"* (Row 2); (3) 27.3% wrong in predicate name of logic component (Row 3). In our opinion, almost the improvement of the PhraseTransformer when compared with Transformer is based on the capacity of capturing local context information.

Table 6: Examples that frequent incorrect predictions of Transformer, are improved in PhraseTransformer on the Atis test set.

| | |
|---|---|
| *Sentence* | *what are the flight **from ci1 to ci2** that **stop in ci0*** |
| *Gold LF* | ( lambda $0 e ( and ( flight $0 ) ( from $0 ci1 ) ( to $0 **ci2** ) ( stop $0 **ci0** ) ) ) |
| *Transformer* | ( lambda $0 e ( and ( flight $0 ) ( from $0 ci1 ) ( to $0 ci0 ) ( stop $0 ci2 ) ) ) |
| *Sentence* | *give me the cheapest **round trip** flight from ci0 to ci1 around mn0 dn0* |
| *Gold LF* | ( argmin $0 ( and ( flight $0 ) ... ( month $0 mn0 ) **( round_trip $0 )** ) ( fare $0 ) ) |
| *Transformer* | ( argmin $0 ( and ( flight $0 ) ... ( month $0 mn0 ) ) ( fare $0 ) ) |
| *Sentence* | *show me the airport **servic by al0*** |
| *Gold LF* | ( lambda $0 e ( and ( airport $0 ) **( services al0 $0 )** ) ) |
| *Transformer* | ( lambda $0 e ( and ( airport $0 ) **( airline $0 al0)** ) ) |

## 5 CONCLUSION

In this paper, we proposed a novel model named PhraseTransformer that can improve the performance of the Transformer in semantic parsing tasks. We enhance Transformer Encoder to improve the representing ability of the detailed meaning of a sentence based on learning the phrase dependencies. In the methods using Neural Network, this model obtains SOTA results on the Atis dataset and achieves a competitive result with the SOTA in other datasets. We also conducted experiments to compare with Transformer and show the improvement of self-attention in PhraseTransformer architecture. In future work, we would like to extract more information about this architecture about the relationship between words or phrases and how to inject prior knowledge to improve it. We believe that this architecture can be widely applied in many problems using sequence to sequence models such as neural machine translation and abstract text summarization.

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

# A    APPENDIX A

## A.1    SELF-ATTENTION VISUALIZATION

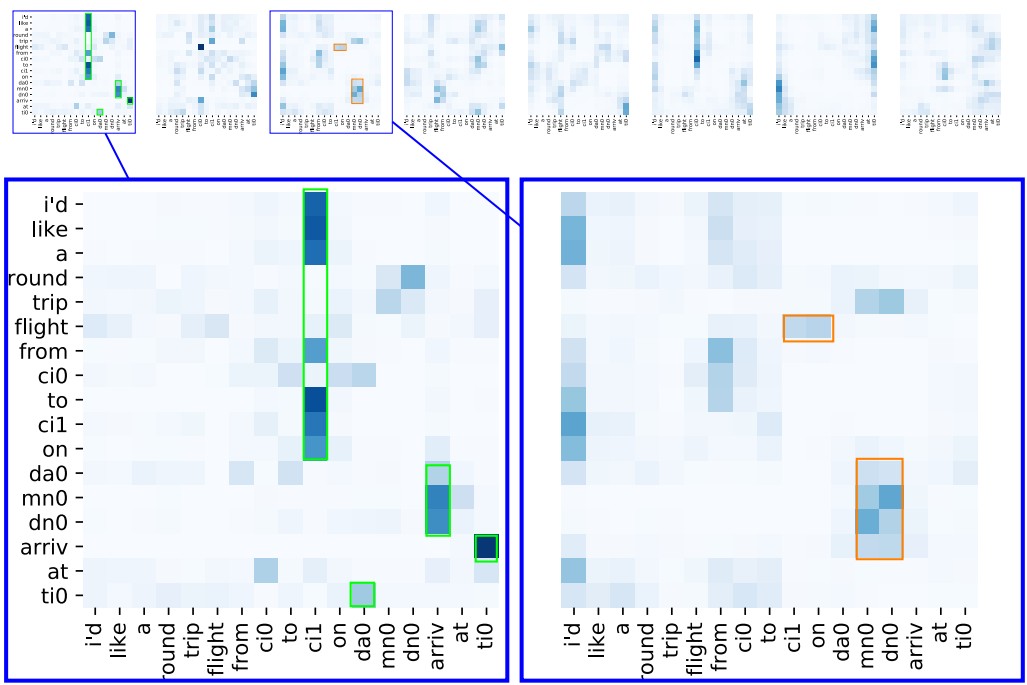

Figure 7: Heatmap visualization of Self-Attention in 8 heads at last layer of PhraseTransformer Encoder. Heads 1 - 8 are ordered from left to right. The gram sizes $n = [0, 0, 2, 2, 3, 3, 4, 4]$. The highlighted rectangles in these heads are highly attended alignments. Two blue rectangles are zoomed-in separately of head 1 (not use $n\_gramLSTM$), head 3 (use $2\_gramLSTM$).

## A.2 ENCODER-DECODER ATTENTION VISUALIZATION

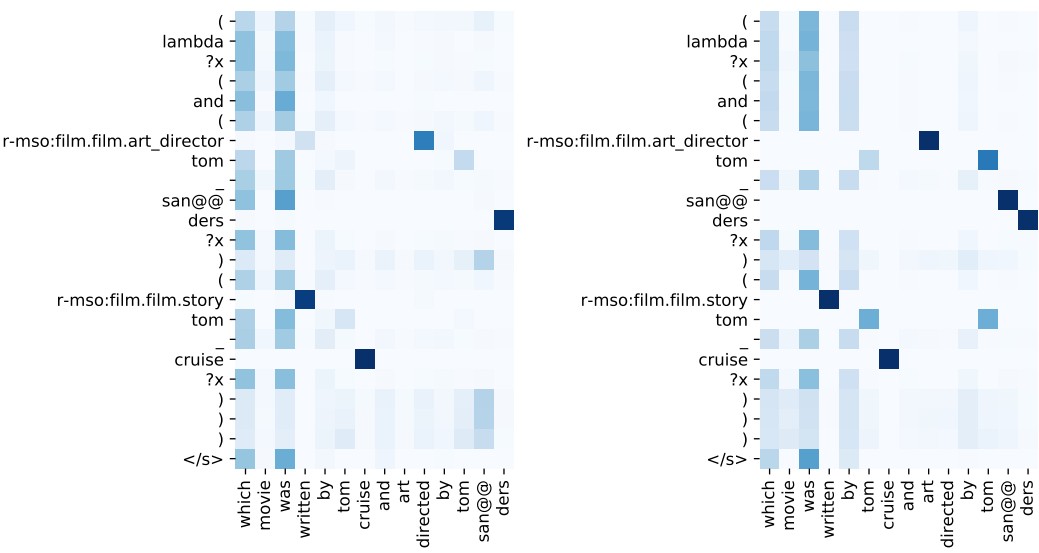

(a) The sentence 6456 in MSParS test set.

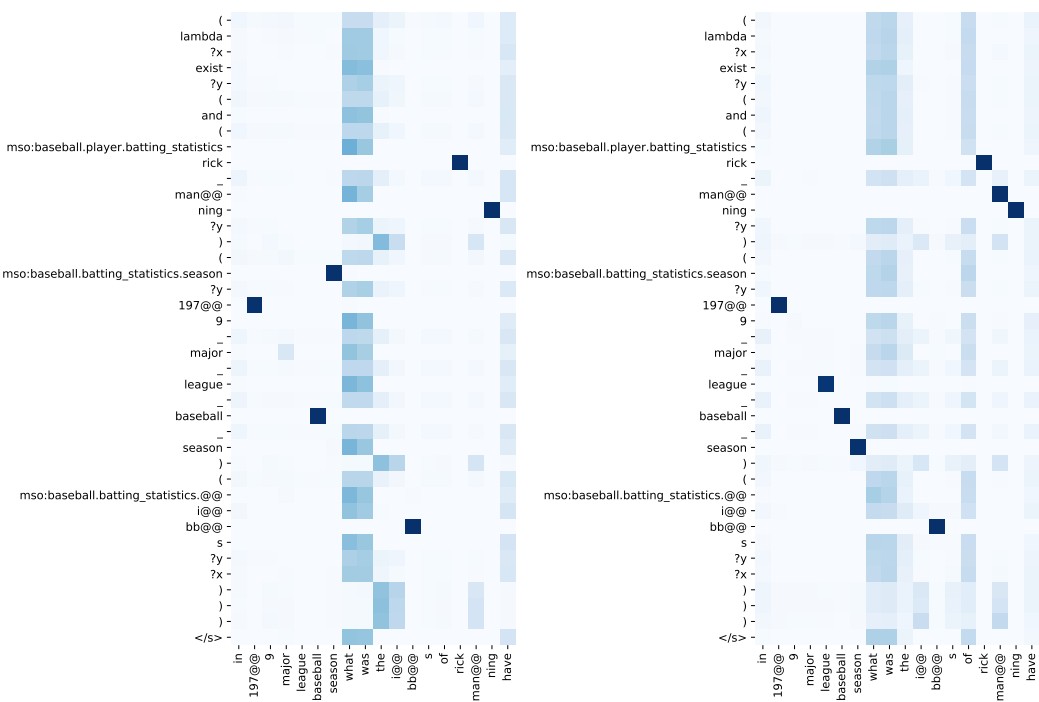

(b) The sentence 440 in MSParS test set.

Figure 8: Heatmap visualization of Encoder-Decoder Attention of Transformer (left) and Phrase-Transformer (right). Two sentences are randomly chosen in the MSParS test set. PhraseTransformer pays more attention than Transformer to words that are entities name in the sentence. For example: words *"tom cruise"* and *"tom san@@ ders"* in sentence 6456 or *"rick man@@ ning"* in sentence 440.

A.3  Similar word vectors by Transformer

In this experiment, we found that the representations of words in the original Transformer is often confusing without considering the local context. Considering two words *"ci0"* (in the context *"from ci1 to ci0"*) and *"ci1"* (in the context *"from ci0 to ci1"*) in Figure 9a, the words closest to these words concentrate on blue and cyan clusters. These clusters are overlapping while PhraseTransformer is separated clearly. Figure 9b shows that the word *"ci0"* in the context *"from ci1 to ci0"* is confused with the *"ci1"* in the context *"from ci0 to ci1"* in many times. The similar problem is showed on the Figure 9c.

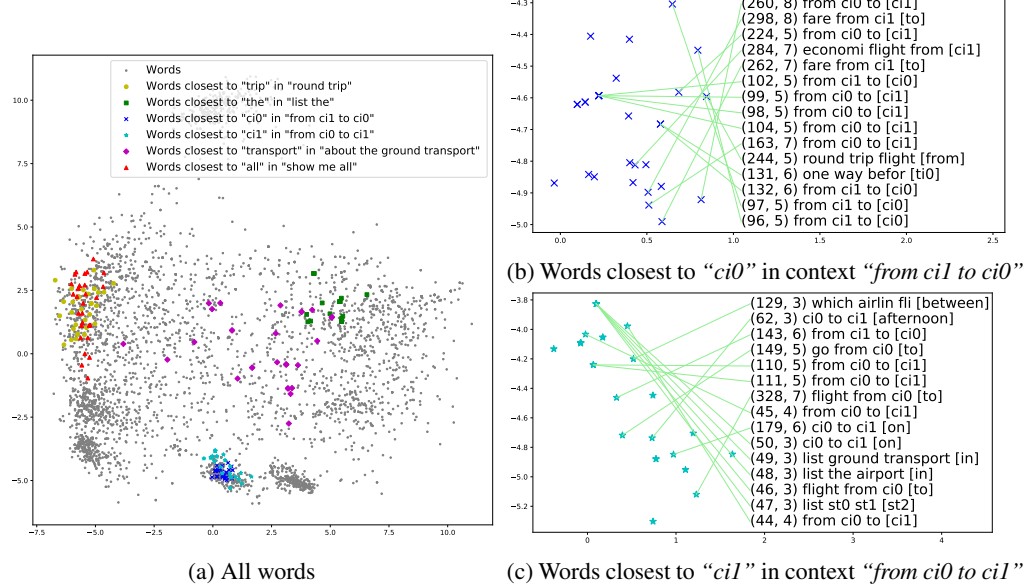

(a) All words

(b) Words closest to *"ci0"* in context *"from ci1 to ci0"*

(c) Words closest to *"ci1"* in context *"from ci0 to ci1"*

Figure 9: Figure a draws the representing vector of words in Selft-Attention Layer using PCA on Atis test set. Figures b, c are zoomed-in view of the blue and cyan clusters. The labels are annotated for each point in two figures show the information of the word corresponding to point following the template *(sentence_id, word_position) phrase_context [considering_word]*.

