# OpenReview forum: "PhraseTransformer: Self-Attention using Local Context for Semantic Parsing"
_ICLR.cc/2021/Conference — Reject_

### Official Review · AnonReviewer4 · 2020-10-25
**novelty is not enough**

**Rating:** 3
**Confidence:** 3

**Review:**

This paper extends Transformer to integrate representations of ngrams for semantic parsing. The key idea is to split input sentences into ngrams of different orders and utilize LSTM to build their representations before feeding them to the layers inside a transformer. The experimental results show that this modification leads to marginal improvement on three benchmark datasets of semantic parsing.

The encoders of the previous neural semantic parsing methods only learn the dependencies between tokens but ignore the local context around the tokens. Therefore, to exploit the local context information, this work provides some main contributions:
-- This work introduced a novel Transformer encoder to encode the n-grams in each utterance.
-- This work showed the effectiveness of this new model on three benchmark datasets.
-- This work displayed the model capacity by visualizing the alignment between the source tokens and the target logical forms, and the similarity of the phrase representations.

The method is evaluated on three datasets, Geo, Atis and MSParS with two evaluation metrics, Exact Matching and Logic Matching. Compared with Exact Matching, Logic Matching is able to compare the variants of the logical forms.

Strengths:

-- It is a good method to exploit the local context information with the Transformer architecture. And this method seems to be easy to implement.

-- There is a thorough evaluation which displays the model performance, and visualizes the attention alignment and the similarity of the phrase representations.

Weakness:

-- The first thing I am concerned about is the novelty. Although in semantic parsing, there is no previous work that utilizes local context information with a Transformer, there are many similar architectures in machine translation. The proposed Transformer-based model is also totally applicable to machine translation scenarios, which makes it necessary to compare the model performance with the machine translation models that exploit local context as well.

-- The second is that the performance is not significant enough. Although this work claims that the performance is superior to the baselines on two benchmark datasets, it should be noted that the other baselines report only the Exact Match accuracy while this work reports the Logic Match performance. Considering only the Exact Match comparison, the proposed method is only 0.4% and 0.02% higher than the baselines on Atis and MSParS, respectively, which is not significant at all. If using Logic Match as the main metric, it would be better to re-evaluate the baselines with Logic Match as well for fair comparison.

--  The description of the model in Sec. 3 is not clear enough.
How does the model split an input sentence into ngrams? Do the adjunct ngrams have overlapped subwords/characters or not?

-- I found this contribution too close to the following work.
Hao, Jie, Xing Wang, Baosong Yang, Longyue Wang, Jinfeng Zhang, and Zhaopeng Tu. "Modeling recurrence for transformer." arXiv preprint arXiv:1904.03092 (2019).

-- In addition:

---- The citation of logic matching seems to be incorrect.

---- The sentence “W is parameters, LayerNorm, FeedForward are the functions by proposed by … .” is misleading because LayerNorm and feed forward networks are not proposed by Vaswani et al.

---- What is model 4 and model 5 in Table 3?

---- Figure 5 (b) is hard to read

Minor issues and improvement suggestions:

-- Compare the proposed method with the baselines which are from machine translation fields which also exploit the local context.
-- Improve the model performance.

Reasons to accept:
-- Proposed an interesting Transformer-based encoder to exploit the local context information.

Reasons to reject:
-- Novelty is not enough. The evaluation results can not show the superiority of this model.

---

> ### Author Response · Authors · 2020-11-24
> **Response to AnonReviewer4**
>
> Thank you for reviewing. About the weakness points:
> 1. **Compare the model performance with the machine translation models.** Thank you for the suggestion.
> 2. **Re-evaluate the baselines with Logic Match as well for fair comparison.** First, we just clear that our baseline - Transformer using Logic Matching in Table 3.  Regarding the previous works except for Coarse2Fine (Dong & Lapata, 2018), we understand your concern, however, it is difficult to get the result for re-evaluate all the previous works. That is the reason we show 2 evaluation Logic Match (LM) and Exact Match. In another aspect, Logic Match just is a method that contains post-processing re-order the logic component, and do an Exact Match evaluation. In the Semantic Parsing task, in our opinion, LM is better to evaluate the performance of the model.
> 3. **How does the model split an input sentence into ngrams?** We answer the way to our proposed model separate sentence to n-grams in Reviewer 3 at question 2.   Yes, the adjunct n-grams have overlapped word/subwords.
> 4. **Contribution too close to the work [1].** To the best of our knowledge, this paper is similar to our work in using RNN incorporating with Transformer but quite different in motivation and architecture. The hypothesis of the author is the recurrence feature is useful for the NMT system, therefore, they proposed an approach to extract and using recurrence features as a new Encoder. Aim to extract, the authors propose a special additional component Attentive Recurrent Network (ARN) into self-attention and show the experimental results to prove the effectiveness of the proposed model. In our work, we explore the local context features created by the phrase in a sentence. We represent the word incorporating localness information by RNN and learn the dependencies between them via the self-attention layer.
>
>   [1] Hao, Jie, Xing Wang, Baosong Yang, Longyue Wang, Jinfeng Zhang, and Zhaopeng Tu. "Modeling recurrence for transformer." arXiv preprint arXiv:1904.03092 (2019).
>
> About additional issues:
> - **The citation of logic matching seems to be incorrect.** Logic Matching, in this case, is not Logic Matching in mathematical logic, we cited Dong & Lapata (2018) because the authors developed this metric and public it with source code from paper.
> - **LayerNorm and feed forward networks are not proposed by Vaswani.** Thank you for the correction. I fixed it in the rebuttal version.
> - **What is model 4 and model 5 in Table 3?** Model 4,5 in Table 3 is referred to the "Id" of the model in Table 2 (we mentioned in the caption of Table 3).
> - **Figure 5 (b) is hard to read.** Thank you for the suggestion. We add a bigger version in Appendix A.1 and refer to it in the rebuttal version.
>
> About minor issues and improvement suggestions, thank you for the suggestions.
>
> In our opinion, our model is simple but effective. Our experiments on datasets show that PhraseTransformer effectively works to understand the complicated in natural language (special in Atis dataset).  Besides, our model is able to apply to the wide application without any explicit additional information.

---

### Official Review · AnonReviewer3 · 2020-10-26
**This paper takes on a challenging task, i.e., semantic parsing from natural language text....**

**Rating:** 7
**Confidence:** 4

**Review:**

##########################################################################
Summary:
This paper takes on a challenging task, i.e., semantic parsing from natural language text.  The paper proposes an improved version of the transformer for the task of semantic parsing, called PhraseTransformer to overcome the limitations (i.e., failure to capture local sentence context effectively) of the transformer. It proposes to use LSTM networks and the Self-Attention mechanism of the original Transformer and provides experimental results for up to 4-grams representations of the sentences to better accommodate the local contexts and achieves state-of-the-art (SOTA) performance on ATIS dataset, and competitive performance on other datasets such as Geo and MSParS.

##########################################################################
Reasons for score:

Overall, I like the idea and I would like the paper to be accepted. The main reason for my acceptance is that in a natural language many values are actually phrases (e.g., one way) and it is very critical to capture the meaning of n-grams as one unit (i.e., one representation for the full n-gram) and consider their context locally to better understand the meaning in the context of the sentence, and effectively parse the natural language texts.

##########################################################################Pros:

1. The paper tackles a very important problem, i.e., parsing natural language text into a semantic frame. Natural language is complex and the semantic frame is easy to handle, which facilitates many down-stream NLP tasks.

2. The proposed approach is also very interesting, i.e., captures the meaning of the phrases or n-grams in the context of the whole sentence.

3. Reasonable experiments have been provided to prove the efficacy of the proposed approach. Moreover, it also outperforms the SOTA model on the ATIS dataset.

##########################################################################
Cons:

1. Although the paper compares with several baselines and SOTA models, an important model for comparison is skipped, i.e., SpanBERT: Improving Pre-training by Representing and Predicting Spans, recently published in ACL 2020 (July 2020 published more than two months ago). Still, I will not recommend rejecting this paper only because of this.

2. The paper fails to explain in detail how n-grams are generated? And how these representations are put back to generate predictions at the word level. Since some n-gram would make sense to form a phrase (e.g., “one way” or “San Francisco”), but many would not be valid phrases (e.g., “a one”, “way ticket”, “ticket from” are not valid phrases in natural language text). Also, it is not trivial to know in advance what is the suitable value for “n” in the n-grams for a given sentence. Please explain this step: not sure but maybe a working example can explain it in a much better way.

##########################################################################
Questions during the rebuttal period:

Point # 1 in the cons section is not very critical for my decision, it may help the authors to improve their paper.

Point # 2 is critical for me to understand the key idea. I would love to see a working example that shows a sentence, how n-grams are generated (also, how n is decided), and how final predictions are performed.

---

> ### Author Response · Authors · 2020-11-24
> **Response to AnonReviewer3**
>
> Thank you for reviewing. About your concern points:
> 1. **Other works comparison.** We updated some comparisons with other works in the rebuttal version.  About SpanBERT, the authors propose (1) a method for pretraining phases: “masking spans” using a geometric distribution and (2) a method to optimize a span-boundary objective.  We see that it is similar to our work about the idea of utilizing useful information from span/phrase for increase understanding capacity of the model but quite a difference in implementation.
> 2. **How n-grams are generated?** In our experiments, on both training and testing phases, we do not use any special method or external tool to separate the phrase or generate the n-gram. We used the overlapped phrase to represent all origin words in a sentence. For example: sentence “a one way ticket from ci0 to ci1”, in the head using 2-gramLSTM, the original word hidden states will be replaced by phrase hidden states (last hidden state of BiLSTM architecture) [(<pad>-a), (a-one), (one-way), (way-ticket), (ticket-from), (from-ci0), (ci0-to), (to-ci1)]. We hypothesize that the local context can be represented by these grams and different gram-size make the different views about the local context (larger gram-size equal to longer local context scope). We agree that there are many invalid phrases, however, the self-attention based on the frequency of the valid phrase to decide what is phrase needs to be attended to. Of course, there is room to improve this mechanism by using the explicit probabilities of phrase to help the model pay attention more effectively, however, the evidence that we show in part (4.3.2) Meaning phrase and Alignment show that the model can learn effectively to represent the meaning of phrases.

---

### Official Review · AnonReviewer1 · 2020-10-27
**interesting directions but more work needs to be done**

**Rating:** 3
**Confidence:** 5

**Review:**

*Summary of the paper*: One drawback of the transformer architecture is that they often fail to capture local interactions within the sentence. In this paper, the authors propose the PhraseTransformer architecture which incorporates Long Short-Term Memory (LSTM) into the Self-Attention mechanism of the original Transformer to capture more local context of phrases. Experimental results on three semantic parsing datasets show that the phrase level transformer is better at capturing the local information than the original transformer.

*Strength of the paper*:  This paper proposes to use phrase-level information to capture the local dependencies of the transformer architecture and the experimental results show that semantic parsing could benefit from such local dependencies. The authors also conduct empirical experiments -- section 4.3.2 to show why their method is helpful. The paper is well-structured and easy to follow.

*Weakness of the paper*:

(1) The idea of using different-granularity representation including phrase-level representations of transformer architecture has been proposed before. For example, Hao, et al. 2019 study multi-granularity representations for self-attention in machine translation; Yang, et al. 2018 study the localness of the self-attention mechanism; Nguyen, et al. 2020 study the tree-structured representations of the self-attention networks. The authors should have done a detailed literature review along this line of works.

(2) The proposed method is quite empirical and from Table 2 it is unclear how to set the n-gram size for different layers. A more detailed experiment demonstrating how different n-gram sizes in different layers affects the model would be quite helpful.

(3) The improvement over the original transformer is marginal except on the Atis dataset, the claims would be more convincing if the authors could conduct similar experiments on other tasks.

(4) For the error analysis part, I would like to see a more systematic analysis rather than two examples posted in the paper: is there a specific type of error being corrected by the phrase-level representation? In what scenario will the phrase-level representation help most?

*Reason for score*: Overall, I vote for rejecting this paper. I like the idea of trying different-granularity representations for the transformer. However, such kind of ideas has been proposed before and this paper does not bring new insights into using such representations.

*Reference*:

Hao, Jie, et al. "Multi-Granularity Self-Attention for Neural Machine Translation." arXiv preprint arXiv:1909.02222 (2019).

Yang, Baosong, et al. "Modeling localness for self-attention networks." arXiv preprint arXiv:1810.10182 (2018).

Nguyen, Xuan-Phi, et al. "Tree-Structured Attention with Hierarchical Accumulation." arXiv preprint arXiv:2002.08046 (2020).

---

> ### Author Response · Authors · 2020-11-24
> **Response to AnonReviewer1**
>
> Thank you for reviewing. About the weaknesses points:
>
> 1. **Detailed literature review along this line of works.** Thank you for the valuable comment. We added a new part in the Related work of the rebuttal version.  About some papers that you mentioned:
>
>   These works [1], [2] focus on incorporating the external tree structure information into the self-attention network while our model does not. Phrase mechanism can be applied for multimedia data (e.g., image) or time-series data. This point makes our model has a strong reproducibility and wide application.
>
>     - About paper [1], the authors add the nodes information (e.g., NP, VP) of the syntactic tree, and the self-attention layers are designed to pay attention to these nodes. However, in Multi-Head layers, the difference between heads is still generated by Linear layers (that is not a natural linguistic feature), while our method feeds different natural linguistic features (different n-grams) for heads.
>     - About paper [2],  it is quite similar to us about the idea that makes difference linguistic features between heads. However, this model replaces the roles of granularity representations in heads. For example, the hidden state of input sequence N words $[h_1, h_2, .. , h_N]$ is replaced by M nonoverlapped phrase or subtree $[p_1, p_2, .. , p_M]$. Intuition, this mechanism will compress information via stacked Encoder layers of the Transformer. Therefore, the model is able to learn the structure information, but the best performance is found when applying only to the first layer. In our method, we keep representations of N words incorporating the local context (contiguous overlapped phrase) $[h’_1, h’_2, .. , h’_N]$ via stacked Encoder layers. Besides, our model is able to keep the original feature of words (in some heads that do not apply n-gram phrase mechanism) and the various type context information (in heads apply different n-gram phrase mechanism), so the hidden state of words in the higher layer contains more context information.
>     - About paper [3], this model is similar to our idea to incorporate the local context but different in localness modeling. In more detail, by casting the localness information as a Gaussian bias before computing self-attention, the model can pay more attention to the local context around considering words. Similar to [1], this architecture does not make differences in linguistic features between heads.  Besides, it is difficult to explain the features learned by the Gaussian component (localness modeling) logically such as what is a good central point, or good window size.  According to the analyzed result, window size (local context scope) in this model quite large with a range from 20 – 50 words while our local context is combined by 2-4 words. Our method is quite simple, clear and our analysis shows the model is able to learn to represent meaning local context effectively.
>
>     *References*
>     - [1] Nguyen, Xuan-Phi, et al. "Tree-Structured Attention with Hierarchical Accumulation." arXiv preprint arXiv:2002.08046 (2020).
>     - [2] Hao, Jie, et al. "Multi-Granularity Self-Attention for Neural Machine Translation." arXiv preprint arXiv:1909.02222 (2019).
>     - [3] Yang, Baosong, et al. "Modeling localness for self-attention networks." arXiv preprint arXiv:1810.10182 (2018).
> 2. **How different n-gram sizes in different layers affect the model.**  We added some new experiments about different n-gram sizes in different layers and presents the part “PhraseTransformer Encoder layers” in part 4.3.1 of the rebuttal version. Besides, the previous answer (comparing with [2]) to us also supports this problem.
> 3. **Conduct similar experiments on other tasks.** Thank you for the suggestion.
> 4. **The error analysis.** We updated the part “Errors examples” -> "Improvement example" in the rebuttal version to make clear about the improvement of PhraseTransformer when compare with the Transformer. In fact, the phrase-level representation helps most in the scenarios that need to have local information to understand. For example sentence “what is the lowest fare **from ap0 to ci0**”, the origin Transformer predicts “...( from \\$1 ci0 ) ( to \\$1 ap0 )..." while the true output is "...( from \\$1 ap0 ) ( to \\$1 ci0 )...". In general, our proposed model works effectively with the paraphrasing problem, which is a critical problem in NLP.
>
> In our opinion, our model is strong in learning the important latent phrase without explicit information. The visualization Encoder-Decoder attention (Figure 5, and Appendix A.2) shows that our model is better at natural language understanding than the original Transformer.

---

### Official Review · AnonReviewer2 · 2020-10-29
**Good motivation, execution and results - but issues with model selection, scalability and impact on the community**

**Rating:** 5
**Confidence:** 5

**Review:**

The paper modifies the self-attention mechanism in transformers to function at the phrase level, rather than at the token level, as a means to improve alignments between input phrases and logical form predicates for Semantic Parsing tasks. They achieve this by using LSTMs on the token representations to form n-gram representations in the attention head, and then performing attention on these n-gram representations. They call this modified transformer, a PhraseTransformer. They are able to improve over baseline transformers on 3 datasets.

**Strengths and reasons to accept**

1. The model improves upon existing neural models on three dataset, achieving SOTA on MSparS.
2. The idea is adequately motivated and the model analyses appropriately corroborate the motivation.
3. The model is described well. Other than the tiny size of the figures and grammatical errors, the model is quite easy to follow.

**Weaknesses and possible reasons to reject**

1. I'm concerned about repeated evaluations on the test set i.e. the model might be overfit to the test set. For example the numbers reported in Table 2, seem to be test set numbers, since they exactly match the numbers in Table 3. This suggests that n-gram sizes were fine-tuned on the test set, which is concerning.
2. How does adding multiple LSTMs to the attention affect the runtime complexity and parallelization ability of the attention mechanism? I feel that the LSTMs add a lot of overhead to the attention mechanism, and might not scale to larger datasets. Can the LSTM be replaced with a simpler operation to achieve the same result?
3. Although they outperform recent neural models, don't forget about Wang et al. 2014 (Morpho-syntactic Lexical Generalization for CCG Semantic Parsing) who are still the SOTA on ATIS with an accuracy of 91.3
4. The datasets used are quite old and performance on these datasets is quite saturated. Might be useful to evaluate on newer semantic parsing datasets such as TOP or other SQL based datasets such as WikiSQL or SPIDER.

**Other issues:**

1. The citation format is quite terrible in the paper. There are no separators whatsoever between the cited papers and the main text.
2. Lots of grammatical errors. Very hard to understand in certain spots.
3. Figure 2 is extremely tiny on print and even on screen.

**Update after rebuttal - I would like to keep my score**
1. Although the authors have provided dev set numbers, the fact that test set numbers were computed, is highly concerning. In fact, for ATIS, the best dev model is different from the best test model. The other models should never have been evaluated on the test set.
2. Am still concerned about the datasets being old and saturated, and would love to see results on more recent datasets.

---

> ### Author Response · Authors · 2020-11-24
> **Response to AnonReviewer2**
>
> Thank you for reviewing. About the minor issues (citation format, grammatical error, figure 2), we fixed them in the rebuttal version.  About the weaknesses points:
> 1. **Model selection.** Actually, we fine-tuned the gram-sizes based on the dev set. To make it clearer, we append the results on the dev set into Table 2 of the rebuttal version.
> 2. This question contains 2 points:
>     1. **Runtime complexity.** Although we used LSTM architecture on Heads, the computation time is not dependent on the length of a sentence because we can forward and backward all $n$-grams of all sentences in a minibatch at the same time (parallel computing). Therefore, the computation time is more dependent on the gram size (in our experiments, the maximum gram size is 4). The training speed of PhraseTransformer is 76-79% of the origin Transformer. We also append a paragraph “Computation time” in 4.3.1 in the rebuttal version.
>     2. **Can the LSTM be replaced with a simpler operation to achieve the same result?** Yes, the LSTM can be replaced with other operations that represent the local context. We tried to replace LSTM with the Sum function and update the result to the rebuttal version. The PhraseTransformer improve slightly performance on the Atis dataset show that our phrase mechanism works effectively. However, in our opinion, LSTM is the potential architecture for representing short text.
> 3. **SOTA on ATIS by Wang et al. 2014.** Thank you, we adjusted some sentences in the rebuttal version for this problem.
> 4. **Useful to evaluate on newer semantic parsing datasets.** Thank you for the suggestion.
>
> In our opinion, the PhraseTranformer model is simple but effective, special in the self-awareness capacity of the model. Besides, it is generalized and can be applied to various tasks in NLP.

---

### Decision · Program_Chairs · 2021-01-07
**Final Decision**

**Decision:**

Reject

**Comment:**

This paper proposes an attention mechanism that works at the phrase level for semantic parsing.
Reviewrs agree that the idea has been previously explored outside semantic parsing, that the gains should be shown on less saturated datasets, and that there are issues in the experimental design (observing test set results for many experiments). Thus, at this point I recommend that the paper is rejected.